# Analysis of astigmatism outcomes after horizontal rectus muscle surgery in patients with intermittent exotropia

**Dong Cheol Lee** [1]*, **Se Youp Lee** [2]

**1** Department of Ophthalmology, School of Medicine, and Institute for Medical Science, Keimyung University, Daegu, Korea, **2** Department of Ophthalmology, School of Medicine, Keimyung University, Daegu, Korea

* tking33@naver.com

**Data Availability Statement:** All relevant data are within the paper and its Supporting Information files.

**Funding:** This work was supported by a research-promoting grant from Keimyung University

## Abstract

This study examined the factors affecting corneal curvature change after lateral rectus recession and medial rectus resection surgery in patients with intermittent exotropia. This was a retrospective cross-sectional study in intermittent exotropia patients who underwent rectus resection surgery. The study involved 41 male and 42 female patients (mean age: 9.55 ± 5.03 years, range: 3–28 years). Corneal astigmatism analysis was performed using the Galilei G4 Dual Scheimpflug Analyzer. The values of simulated and ray tracing corneal keratometry (K) of astigmatism, including axis changes, were determined preoperatively and at 1 week and 3 months postoperatively. The factors found to affect corneal curvature change were sex, extent of surgery, and axial length. Simulated and ray tracing changes were significant preoperatively and at 1 week and 3 months after rectus resection surgery ($p$ < 0.05); however, there were no differences in astigmatism (D) at any time. The spherical equivalent had a myopic change after rectus resection surgery with cycloplegic refraction, and in ray tracing mode, flat K was decreased at 1 week from baseline and increased 3 months later. Steep, mean K, and axis increased continuously from baseline to 1 week and 3 months. Astigmatism, in contrast, was increased at 1 week, but decreased at 3 months, with no return to baseline. Univariable linear regression analyses showed that the extent of surgery had an effect on flat K change and that sex had an effect on steep K and axis. Additionally, axial length affected steep K and astigmatism, while age had no effect on any variable. Ray tracing values were significantly different from simulated values. In ray tracing mode, rectus resection surgery may result in astigmatism shifted toward with-the-rule, and myopic changes may be caused by differences in thickness and flexibility of the sclera. Notably, age did not affect any variable.

## Introduction

Strabismus is a well-known ocular disorder with a prevalence of 3.8–5% [1]. Surgery is a major treatment of choice for strabismus. Surgical correction to reposition the ocular muscle by weakening the lateral rectus muscle and strengthening the medial rectus muscle, thus

Dongsan Medical Center in 2018. No funding was received for the conduction of this study.

**Competing interests:** The authors have declared that no competing interests exist.

alternating the orientation of their actions, has been the treatment of choice for exotropia [2]. In particular, intermittent exotropia (IXT) is the most common type of strabismus in South Korea, with an overall incidence of 1.1 ± 0.1%, according to the Korea National Health and Nutrition Examination Survey data [3]. Previous studies have shown postoperative refractive error changes and visual disturbances after horizontal muscle surgery, such as myopic and hyperopic shifts, or shifts toward with-the-rule astigmatism [4–10]. Nonetheless, it has been found that, in patients who undergo surgery for strabismus, the refractive error changes are transient [4, 11, 12] and not statistically significant [4, 13]. Most previous reports on strabismus astigmatism have employed simulated keratometry (K) [4–13]. However, it is important to highlight that simulated K does not correct the post corneal curvature, as it only corrects the anterior corneal curvature.

Recently, various imaging techniques have been developed for corneal curvature assessment, including corneal topography devices combining a Placido disc and a Scheimpflug camera. While two cameras are rotated to measure the anterior segment, it is known that this measure reduces the error due to the deviation from the center [14]. The dual-camera system derives images from both sides, which minimizes the effect of decentration due to eye movements on posterior corneal curvature and corneal pachymetry measurements [15]. This device can correct the posterior as well as the anterior corneal curvature. Importantly, in pediatric patients in particular, it is not possible to identify various symptoms accurately. Moreover, the symptoms may overlap with blurred vision due to diplopia. Therefore, it is crucial to pay attention to such postoperative refractive changes, especially in children who are prone to recurrence of deviation and amblyopia. Furthermore, sex, age, extent of surgery, and axial length seem to affect the postoperative corneal changes. Axial length, in particular, is one of the most important factors affecting refractive error, and the progress of myopia is related to the increase in axial length [16].

The purpose of this study was to evaluate the postoperative real astigmatism changes using the Galilei G4 Dual Scheimpflug Analyzer (Ziemer, Port, Switzerland), not only in the anterior corneal curvature but in the posterior curvature as well, and to investigate various risk factors (sex, age, extent of surgery, and axial length) affecting postoperative corneal curvature changes in exotropia patients who underwent rectus resection surgery (R&R).

## Materials and methods

### Patient demographics

The design of the present study followed the tenets of the Declaration of Helsinki for biomedical research in human subjects and was approved by the institutional review board (IRB No. 2016-11-004) of Keimyoung University Dongsan Medical Center. Informed consent was not required as this was a retrospective study, and the data were analyzed anonymously. We retrospectively reviewed the medical records of patients who had undergone lateral rectus recession and R&R for intermittent exotropia from June 2015 to February 2016.

Eighty-three eyes of 83 patients, who underwent R&R for correction of exotropia, were enrolled in this study. Forty-two patients (50.6%) were female, and the mean age was 9.55 ± 5.03 years (range: 3–28 years). Subjects were divided into two groups (below 8 years old and 8 years or older). In addition, the average axial length of patients was 23.30 ± 1.32 mm. Axial lengths (AL) were divided into two groups (less than 23.15 mm and 23.15 mm or longer) (Table 1). We used the median as a measure of central tendency and divided our patients into two groups based on the median value, rather than the average. We then conducted our analysis based on this classification.

**Table 1. Demographics of the study population.**

| General characteristic | Preoperative |
|---|---|
| Mean age (y) | 9.55 ± 5.032 |
| Age (below 8 years: 8 years or older) | 32:51 |
| Male: Female | 41:42 |
| Axial length (mm) | 23.30 ± 1.32 |
| Axial length (less than 23.15 mm: 23.15 mm or longer) | 42:39 (data missing for 2 patients) |
| Extent of surgery ([5.0 mm/4.0 mm] + [6.0 mm/5.0 mm]: above) | 34:49 |

Values are presented as mean ± SD.

Exclusion criteria were patients with previous ocular or neurological diseases, history of ocular surgery, and any vertical deviation. Patients who did not follow the scheduled visits after surgery or who showed poor cooperation in a refractive assessment (autorefractometer) and a prism cover test were also excluded.

## Surgery analytical method

All patients underwent an ophthalmologic examination prior to their R&R surgery. We collected the following information from the preoperative records of patients: age, sex, axial length, mean angle of exodeviation at distance and near, extent of surgery, best-corrected visual acuity, refractive error, and slit-lamp examination results. Deviation angles were measured using the alternate prism cover test at distance (6 m) and near (33 cm) fixation in all nine positions of gaze using accommodative targets and the patients' best optical correction. Corneal astigmatism analysis was performed by the same experienced examiner using the Galilei G4 Dual Scheimpflug Analyzer (Ziemer, Port, Switzerland). Furthermore, SE was evaluated after the prism cover test by a single examiner (LSY) using cycloplegic refraction. See Table 2. These analyses were performed before surgery and 1 week and 3 months after surgery. The corneal topographic measurements determined from the Scheimpflug images were divided into two groups corresponding to simulated and ray tracing mode keratometry. Anterior and posterior astigmatism, including keratometry, were calibrated in ray tracing mode. Anterior astigmatism was only

**Table 2. Mean and standard deviation in ray tracing and simulated mode at each time point.**

| Variables | | Preoperative | Postoperative week 1 | Postoperative month 3 |
|---|---|---|---|---|
| Ray Tracing | Flat K (D) | 41.388 ± 1.553 | 41.209 ± 1.564 | 41.975 ± 1.576 |
| | Steep K (D) | 42.767 ± 1.743 | 43.249 ± 1.776 | 43.8169 ± 1.720 |
| | Mean K (D) | 42.072 ± 1.583 | 42.237 ± 1.606 | 42.907 ± 1.604 |
| | Astigmatism (D) | 1.379 ± 0.958 | 2.110 ± 0.913 | 1.652 ± 0.830 |
| | Axis (˚) | 86.386 ± 16.234 | 90.048 ± 12.406 | 91.819 ± 10.528 |
| Simulated | Flat K (D) | 42.531 ± 1.919 | 42.255 ± 1.543 | 42.340 ± 1.463 |
| | Steep K (D) | 43.829 ± 1.777 | 44.356 ± 1.727 | 44.002 ± 1.690 |
| | Mean K (D) | 43.126 ± 1.590 | 43.301 ± 1.571 | 43.175 ± 1.534 |
| | Astigmatism (D) | 1.4281 ± 0.9434 | 2.101 ± 0.862 | 1.663 ± 0.822 |
| | Axis (˚) | 87.169 ± 13.936 | 87.928 ± 13.675 | 88.964 ± 12.404 |
| CR | SE (D) | -1.328 ± 2.363 | -2.408 ± 2.290 | -1.982 ± 2.161 |

Values are presented as mean ± SD.

CR: cycloplegic refraction, K: keratometry, *SE: spherical equivalent.

compared in simulated mode. All surgeries were performed by the same surgeon (LSY), who performed a limbal incision at the medial rectus muscle resection and an incision in the fornix in the lateral rectus muscle recession (R&R) surgery using six classification groups according to preoperative exodeviation (Table 3). Furthermore, the surgery groups were divided into two groups (recession: resection; 5.0 mm: 4.0 mm + 6.0 mm: 5.0 mm and above). All relevant data can be found in the supporting information (S1 Appendix).

### Statistical analysis

Data were analyzed using the SPSS software version 25.0 (SPSS Inc., Chicago, IL). Most variables are expressed as mean ± standard deviations (SD). The difference between ray tracing and simulated mode at each time point was compared using Student $t$ tests. Paired $t$ tests were used for analyzing the difference in values before and after surgery in simulated and ray tracing mode. Further, a univariable linear regression was used to discern the factors affecting each variable. For all tests, $p$ values less than 0.05 were considered statistically significant.

## Results

Flat, steep and mean K, astigmatism, and axis were analyzed in ray tracing and simulated mode at the initial visit, 1 week after operation, and 3 months after operation. At the same time, the spherical equivalent (SE) was determined by cycloplegic refraction. The results are expressed as mean ± SD as follows. At the initial visit, in ray tracing mode, flat, steep, and mean K (D) were 41.39 ± 1.56, 42.77 ± 1.74, and 42.07 ± 1.58, respectively. Additionally, astigmatism (D), axis (˚), and SE (D) were 1.38 ± 0.96, 86.38 ± 16.23, and -1.33 ± 2.36, respectively. In simulated mode, flat, steep, and mean K (D) were 42.53 ± 1.92, 43.83 ± 1.78, and 43.13 ± 1.59, respectively. Lastly, astigmatism (D) and axis (˚) were 1.43 ± 0.94 and 87.17 ± 13.94, respectively (Table 2).

The difference between the values of ray tracing and simulated mode at each time point (initial visit, 1 week after operation, and 3 months after operation) is expressed as the mean and standard deviation, and the statistical significance of the results was examined. The differences in flat K (D) between the two modes for the three time points mentioned above were as follows; -1.14 ± 0.27 ($p < 0.001$), -1.05 ± 0.24 ($p < 0.001$), and -0.37 ± 0.24 ($p = 0.123$). In steep K (D), we found differences of -1.06 ± 0.27 ($p < 0.001$), -1.11 ± 0.27 ($p < 0.001$), and -0.19 ± 0.27 ($p = 0.485$). In mean K (D), the differences found were -1.05 ± 0.25 ($p < 0.001$), -1.06 ± 0.25 ($p < 0.001$), and -0.27 ± 0.24 ($p = 0.273$). The differences in astigmatism (D) were -0.05 ± 0.15 ($p = 0.740$), +0.01 ± 0.14 ($p = 0.952$), and -0.01 ± 0.13 ($p = 0.929$). Finally, the differences in axis (˚) were -0.78 ± 2.40 ($p = 0.739$), +2.12 ± 2.03 ($p = 0.297$), and +2.86 ± 1.79 ($p = 0.112$) (Figs 1–6).

**Table 3. Classification of extent of surgery by exodeviation prism diopter.**

| Exodeviation angle | Extent of surgery (LR recession/MR resection) | Number of patients |
|---|---|---|
| 20 PD | 5.0 mm/4.0 mm | 13 |
| 25 PD | 6.0 mm/5.0 mm | 36 |
| 30 PD | 7.0 mm/5.5 mm | 17 |
| 35 PD | 7.5 mm/6.0 mm | 10 |
| 40 PD | 8.0 mm/6.5 mm | 6 |
| 50 PD | 10.0 mm/7.0 mm | 1 |
| Total | | 83 |

LR: lateral rectus; MR: medial rectus; PD: prism diopter.

In ray tracing mode, flat K (D) decreased at 1 week from baseline (-0.1783, $p < 0.05$), but it increased beyond the baseline level 3 months later (+0.5871, $p < 0.05$). In steep, mean K (D), and axis (˚) increased continuously from baseline at 1 week (+0.4822 and + 0.1649; $p < 0.05$ and $p < 0.05$) and 3 months (+1.0502 and +0.8345; $p > 0.05$ and $p < 0.05$). In contrast, astigmatism (D) increased at 1 week from baseline, but decreased at 3 months, with no return to the baseline. The SE (D) shifted to myopic change after R&R surgery at 1 week (-1.0727, $p < 0.05$) and 3 months (-0.6537, $p < 0.05$) from baseline (Figs 1–6).

In the univariable linear regression analyses, flat K (D) was affected by even a small extent of surgery, with a significantly higher amount of change 1 week after the surgery (0.304, $p < 0.05$) than before the surgery. In steep K (D), the one-week postoperative change in the three-month post-surgical period was -0.460 ($p < 0.01$) in males and 0.406 ($p < 0.05$) in the AL of 23.15 mm

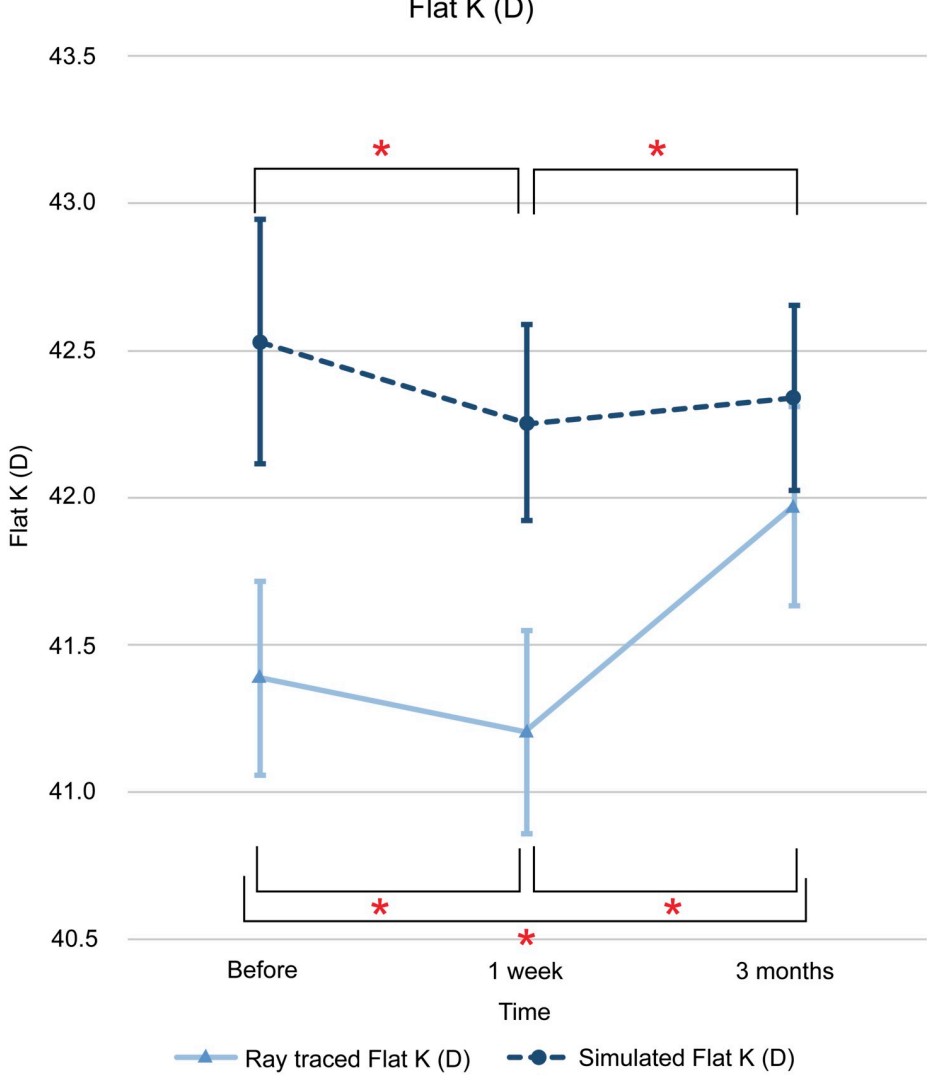

**Fig 1. Comparison of changes in flat keratometric (D) values between ray tracing and simulated mode.** D: diopter.
* $p < 0.05$ by paired *t* test.

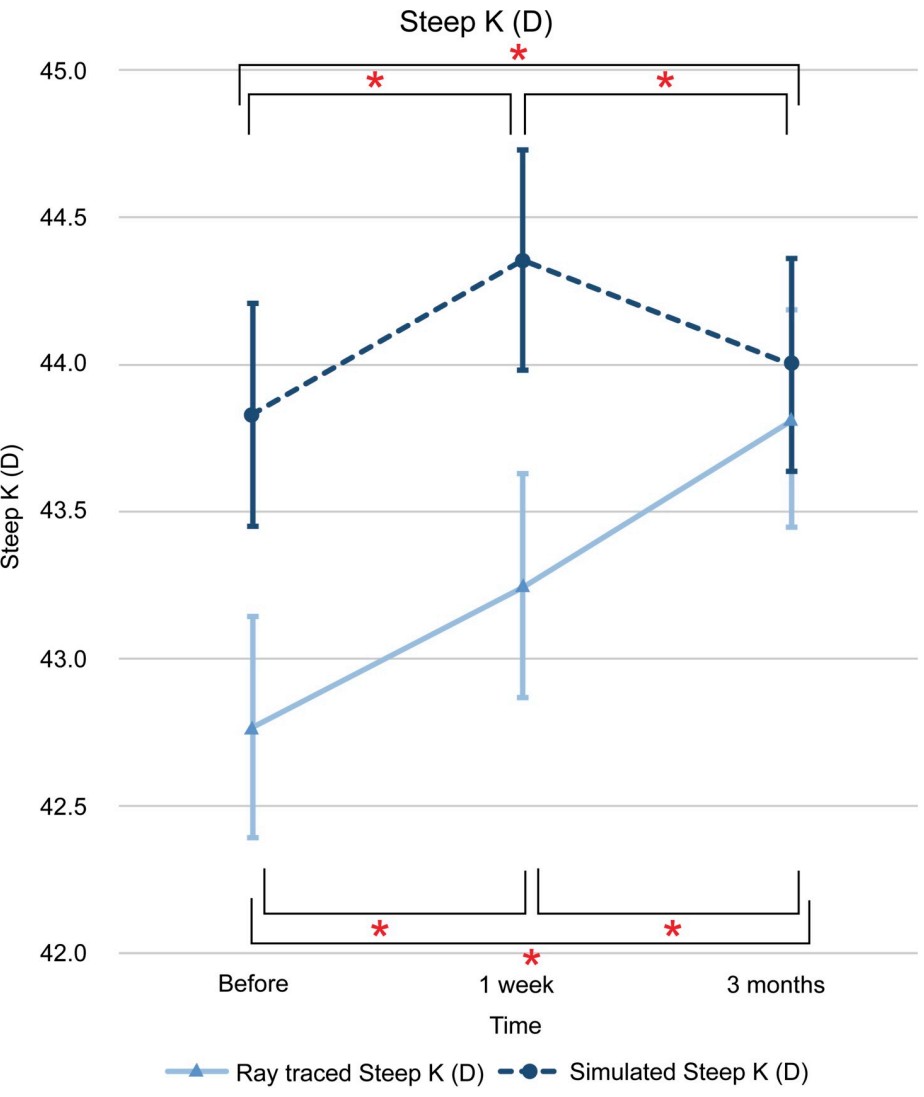

D: Diopter

* p < 0.05 by Paired T test

**Fig 2. Comparison of changes in steep keratometric (D) values between ray tracing and simulated mode.** D: diopter. * $p < 0.05$ by paired $t$ test.

or longer. There were no statistically significant changes in mean K (D) at any time for any variable. In astigmatism (D), with an AL of 23.15 mm or more, the amount of change in astigmatism (D) one week after surgery differed by -0.552 ($p < 0.01$), which was a greater difference than that found before surgery. Regarding the axis (˚), males were affected at postoperative week 1 and 3 months later, with values of -11.351 ($p < 0.01$) and 9.222 ($p < 0.01$), respectively (Table 4).

## Discussion

The results of the present study showed that, except for astigmatism, ray tracing K changes were different from simulated K changes after R&R in IXT patients preoperatively and 1 week after the operation. Therefore, we used ray tracing mode and found that flat K and SE were decreased at 1 week and then increased at 3 months after operation. In contrast, astigmatism

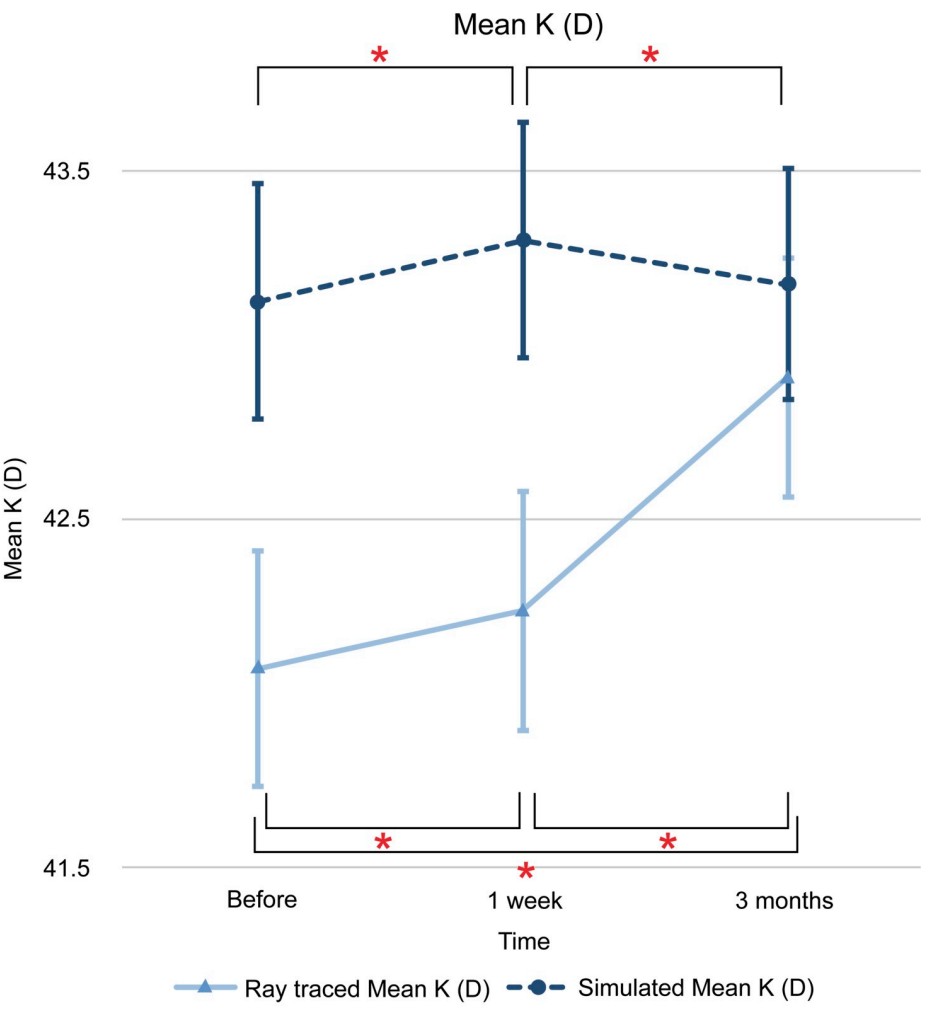

**Fig 3. Comparison in changes in mean keratometric (D) values between ray tracing and simulated mode.** D: diopter. * $p < 0.05$ by paired *t* test.

values were increased at 1 week and decreased at 3 months after operation. Regarding steep, the mean K and axis showed a continuous increase. The univariable linear regression analyses showed that the extent of surgery had a significant influence on flat K change, and sex had an effect on steep K and axis. In addition, axial length affected steep K and astigmatism, while age did not affect any variable.

Previous reports have revealed that 60% of patients undergoing strabismus surgery exhibit a change in astigmatism [4]. In addition, other studies have shown changes in refraction caused by surgery of the extraocular muscles [4, 9, 10, 12, 17]. Most changes of this kind have been thought to be related to changes in the corneal curvature secondary to the reduction in the tension of the recessed extraocular muscle, which is transmitted via the sclera to the cornea [6, 11, 18–21]. A high incidence of increased with-the-rule astigmatism of up to 2 D of change after horizontal muscle R&R surgery has also been reported [7]. Most prior studies have obtained results of corneal changes by using simulated K.

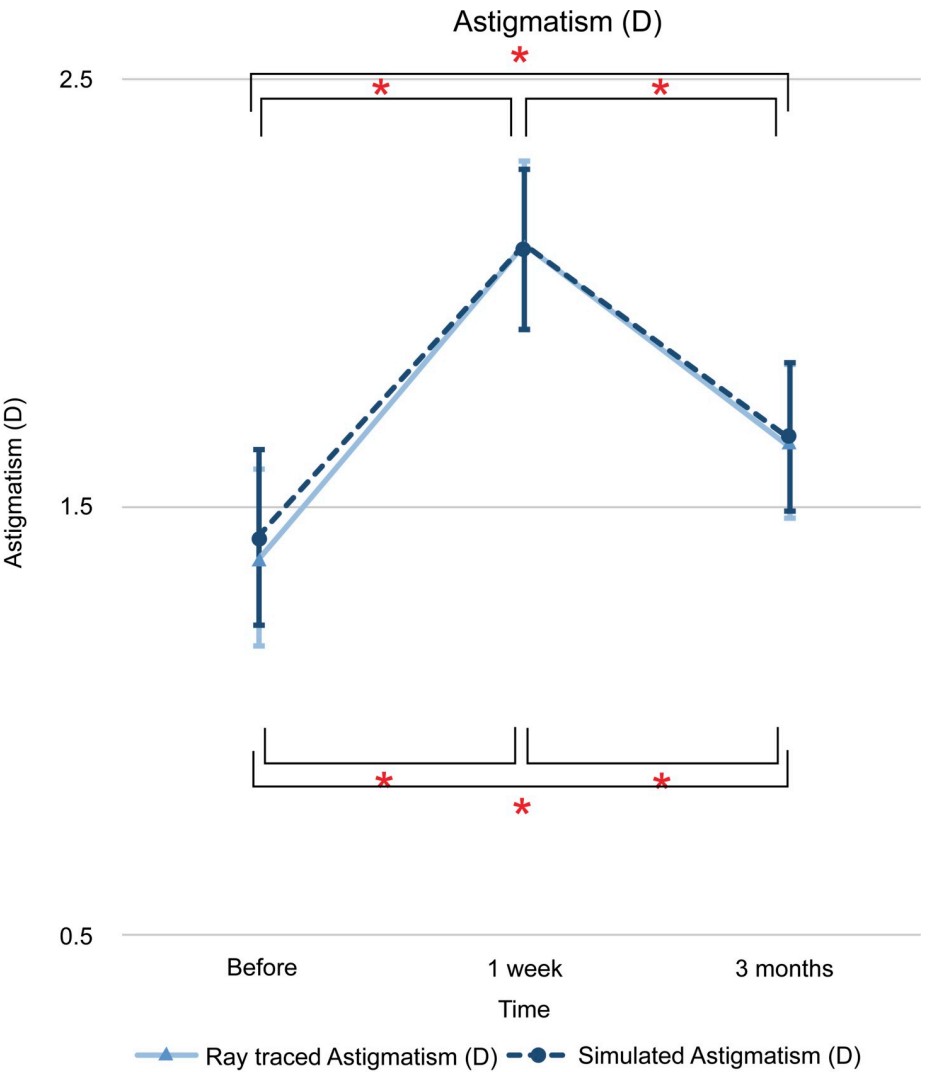

**Fig 4. Comparison of changes in astigmatism keratometric (D) readings between ray tracing and simulated mode.**
D: diopter. * $p < 0.05$ by paired $t$ test.

Modern vision research has benefited from wavefront technology, as it allows the measurement of low and high order aberrations as well as the correction of posterior corneal astigmatisms and the total corneal curvature via the Galilei G4 Dual Scheimpflug Analyzer (Ziemer, Port, Switzerland). Ray tracing using Snell's law and pachymetry data with a reference plane in the posterior corneal surface has also been used [22]. Given that we employed the Galilei G4 Dual Scheimpflug Analyzer to correct both anterior and posterior corneal astigmatisms, the present study may more accurately reflect topography changes after R&R surgery than the simulated K approaches used previously. Furthermore, our study showed that ray tracing K changes were different from Sim K changes after R&R in patients with IXT (Figs 1–6). Therefore, we used the ray tracing mode to investigate the changes in the cornea curvature after surgery in the present study.

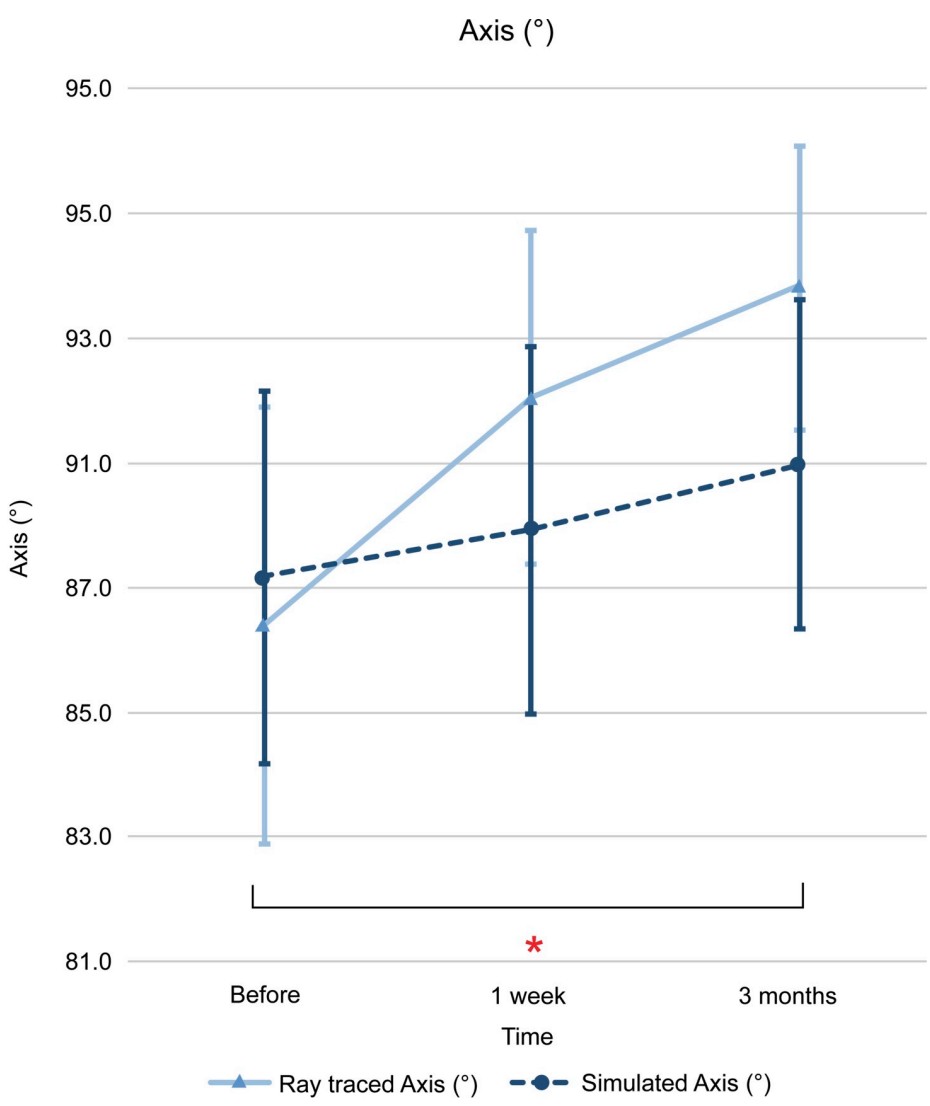

**Fig 5. Comparison of changes in axis (˚) values between ray tracing and simulated mode.** * $p < 0.05$ by paired $t$ test.

Using ray tracing mode, we found that flat K and SE were decreased at 1 week and at 3 months after the operation; conversely, flat K, but not SE, was increased in relation to the baseline. Flat K seemed to be affected by surgery as soon as 1 week after the operation, but then returned to baseline levels or higher. SE showed a myopic shift 1 week after the operation. After that, the myopia slightly decreased, but a myopic shift remained 3 months after the operation. Using an autorefractor, Hong et al. [8] found that R&R surgery continuously affected SE with myopic shifts until 3 months later. This difference was caused using the Galilei G4 Dual Scheimpflug Analyzer as ray tracing mode for correcting posterior corneal values. In contrast, astigmatism values were increased at 1 week and decreased at 3 months after operation. Astigmatism may be changed by R&R surgery with effects on sclera flexibility. As for the steep, mean K and axis continuously increased. This means that R&R surgery affected steep, mean K, and axis not immediately but continuously until 3 months after surgery. Previous reports have shown that these facts might be related to changes in the corneal curvature secondary to the

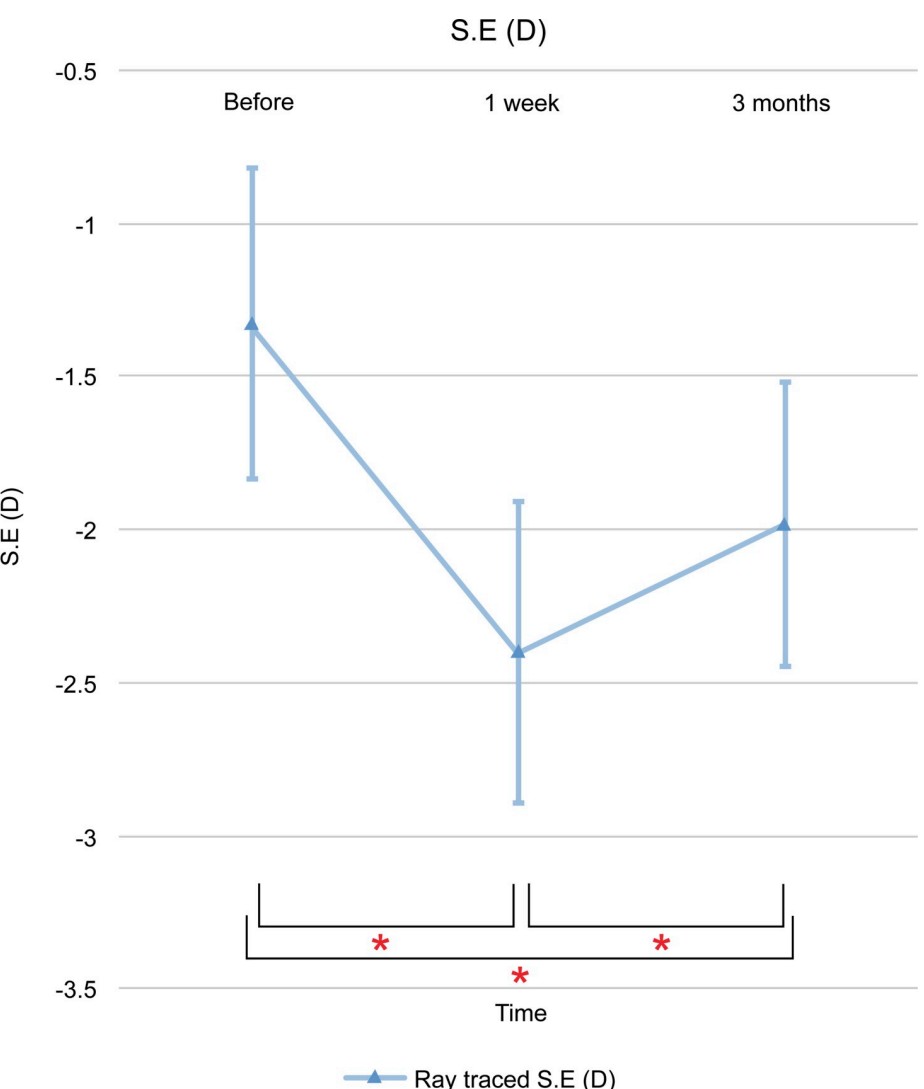

SE: Spherical equivalent, D: Diopter

✳ p < 0.05 by Paired T test

**Fig 6. Comparison of changes in SE (D) values in ray tracing mode.** D: diopter. SE: spherical equivalent. $^*$ $p < 0.05$ by paired $t$ test.

reduction in the tension of the recessed extraocular muscle transmitted via the sclera to the cornea [6, 11, 18–21].

Together, our results of astigmatism measurements are in agreement with previous studies reporting that recession or recession/resection of a horizontal muscle causes changes toward the with-the-rule direction [4–7, 9–13, 18, 19, 21] in ray tracing mode. In addition, some authors have reported that resection has a much smaller effect on induced astigmatism than does recession [10, 19]. These facts from previous reports might also reflect the with-the-rule astigmatism change observed in the present study after R&R surgery. In addition, we investigated how the preoperative axis affected the postoperative change in astigmatism. The preoperative axis was divided by the median value, and the resultant groups were compared. In both the ray tracing mode and simulated mode, the median preoperative axis value was 90 degrees,

**Table 4. Univariable regression of each variable in ray tracing mode.**

| Variables | Factors | Sex | Age | Amount of surgery | Axial length |
|---|---|---|---|---|---|
| Flat K (D) | Postop 1W - Preop | -0.051 | 0.232 | 0.304* | 0.236 |
| | Postop 3M - Preop | -0.130 | 0.089 | 0.031 | 0.109 |
| | Postop 3M –Postop 1W | -0.079 | -0.143 | -0.273 | -0.127 |
| Steep K (D) | Postop 1W - Preop | 0.158 | 0.093 | 0.288 | -0.310 |
| | Postop 3M - Preop | -0.302 | -0.076 | 0.324 | 0.096 |
| | Postop 3M –Postop 1W | -0.460** | -0.169 | 0.035 | 0.406* |
| Mean K (D) | Postop 1W - Preop | 0.022 | 0.210 | 0.252 | -0.015 |
| | Postop 3M - Preop | -0.178 | -0.014 | 0.207 | 0.079 |
| | Postop 3M –Postop 1W | -0.200 | -0.224 | -0.046 | 0.094 |
| Astigmatism (D) | Postop 1W - Preop | 0.046 | -0.170 | -0.157 | -0.552** |
| | Postop 3M - Preop | -0.030 | -0.215 | 0.088 | -0.120 |
| | Postop 3M –Postop 1W | -0.076 | -0.046 | 0.245 | 0.402 |
| Axis (˚) | Postop 1W - Preop | -11.351** | -9.246 | 5.908 | 9.060 |
| | Postop 3M - Preop | -2.129 | -9.757 | 2.522 | 8.890 |
| | Postop 3M –Postop 1W | 9.222** | -0.511 | -3.386 | -0.170 |
| SE (D) | Postop 1W - Preop | 0.045 | 0.213 | -0.068 | 0.290 |
| | Postop 3M - Preop | -0.140 | 0.423 | -0.198 | 0.278 |
| | Postop 3M –Postop 1W | -0.185 | 0.210 | -0.129 | -0.012 |

*$p < 0.05$,

**$p < 0.01$.

and the change in astigmatism in the ray tracing mode increased 1 week after surgery in both the group with preoperative axis values less than 90 degrees and the group with 90 degrees or more. It decreased by 3 months, but there was a difference at 3 months after surgery. Among the subjects whose axis was less than 90 degrees, astigmatism returned to baseline—without achieving statistical significance by paired t test—3 months after surgery. In the group with axis values of more than 90 degrees, the change in astigmatism remained statistically significant 3 months after surgery. The same result was obtained in simulated mode. Future research should study the preoperative axis affected by the vector force due to R&R surgery as well as the effects on the sclera and cornea, in greater detail.

The SE results revealed a myopic shift 1 week and 3 months after the operation, together with a slight hyperopic change occurring from week 1 to month 3 after the operation.

The observation of the myopic shift after surgery returning nearly to its preoperative value at postoperative month 3 is similar to a transient myopic shift reported in a previous study [6]. The myopic shift that remained 3 months after the operation may be explained by the fact that the patients who underwent muscle resection and recession showed higher scleral effects in comparison with the effects observed in nonsurgical patients, whose mean age was 9.55 years. This age coincides with the age at which myopia increases physiologically [23–25].

The univariable linear regression analyses (see above in this section) revealed that age did not affect any variable at any time interval. We also confirmed the significant changes in scleral thickness with age, from childhood into early adulthood, reported in previous works. These changes were more prominently observed in locations distal from the scleral spur [26]. In our report, although age ranged from 3 to 28 years old, with a mean and SD of 9.55 ± 5.03 years old and a median of 8 years old, patients were mostly between 5 and 10 years old. The age distribution was fairly narrow; thus, in our study, it was not possible to determine if the effects of

age were biased. Furthermore, to avoid bias, future studies will have to include wider ranges of age groups from R&R surgery patients. This would allow to compare the effect of surgery among several age groups.

With a small amount of surgery (recession: resection; 5.0 mm: 4.0 mm + 6.0: 5.0 mm), the amount of change 1 week after surgery changed by 0.304 D ($p<0.05$); this was more than the change observed before surgery in flat K only. Others have reported no correlation between changes in the extent of surgery and refractive power [13, 27]. Denis et al. [9] reported that changes in recession and astigmatism were inversely related. In addition, Hainsworth et al. [10] and Kwito et al. [19] have shown that resection has much lesser effect on induced astigmatism than does recession. Based on these facts, a small extent of surgery may be less affected by recession than by resection, immediately increasing flat K after surgery.

Regarding the axial length, values of 23.15 mm or more had a stronger effect on the steep K change from 1 week after the operation to 3 months after operation and resulted in less mean K change from the preoperative visit to 1 week after the operation. Previous studies have reported reduced thickness of the posterior sclera associated with myopia and axial length [28, 29]. According to these previous reports, long ALs may lead to a reduction in scleral thickness and flexibility. These facts may also reduce mean K change immediately after operation and increase steep K changes at the healing stage, 3 months after the operation by recession of the muscle.

Finally, regarding sex differences, male sex had a lesser effect on steep K and a lesser immediate effect but had a higher effect on axis from 1 week to 3 months and in the healing stage. A thicker anterior sclera in males has been reported in a number of previous in vivo studies using a variety of measurement techniques [30–32]. These facts may also be affected by sex differences in thickness and flexibility of the sclera and cause a difference in K change.

The strength of this study is that it directly shows the difference between the change in the anterior cornea, and the correction of the anterior and posterior cornea, before and after surgery. The study also shows the real astigmatism changes in the cornea with the values corrected to the posterior of the cornea in R&R surgery patients using various risk factors.

The limitations of this study include, first of all, its retrospective nature and a short follow-up period of 3 months after operation. Future prospective studies with longer follow-up periods after surgery may provide additional insights into the extent of corneal real astigmatic changes due to R&R surgery. Furthermore, the small sample size, which makes it difficult to generalize our results to a wider population, was another limitation of the present study. A larger sample size, including patients with or without R&R surgery, would enable additional comparisons between control and surgery groups to further investigate corneal changes in IXT patients.

In conclusion, ray tracing values were significantly different from simulated values. In ray tracing mode, R&R surgery may result in astigmatism changes toward with-the-rule. Myopic changes may be caused by differences in the thickness and flexibility of the sclera. Notably, age did not affect any variable.

## Supporting information

**S1 Appendix. Study data.**
(XLSX)

**S1 File.**
(XLSX)

## Author Contributions

**Conceptualization:** Dong Cheol Lee, Se Youp Lee.

**Data curation:** Dong Cheol Lee.

**Formal analysis:** Dong Cheol Lee.

**Funding acquisition:** Dong Cheol Lee, Se Youp Lee.

**Investigation:** Dong Cheol Lee, Se Youp Lee.

**Methodology:** Dong Cheol Lee, Se Youp Lee.

**Project administration:** Dong Cheol Lee.

**Resources:** Dong Cheol Lee, Se Youp Lee.

**Software:** Dong Cheol Lee.

**Supervision:** Dong Cheol Lee, Se Youp Lee.

**Validation:** Dong Cheol Lee, Se Youp Lee.

**Visualization:** Dong Cheol Lee, Se Youp Lee.

**Writing – original draft:** Dong Cheol Lee.

**Writing – review & editing:** Dong Cheol Lee.

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
