## [Decision Letter · Decision Letter 0]

18 Aug 2020

PONE-D-20-20795

Analysis of astigmatism outcomes after horizontal rectus muscle surgery in subjects with intermittent exotropia

PLOS ONE

Dear Dr. Lee,

Thank you for submitting your manuscript to PLOS ONE. After careful consideration, we feel that it has merit but does not fully meet PLOS ONE’s publication criteria as it currently stands. Therefore, we invite you to submit a revised version of the manuscript that addresses the points raised during the review process.

We look forward to receiving your revised manuscript.

Kind regards,

Ahmed Awadein, MD, Ph.D, FRCS

Academic Editor

PLOS ONE

Journal Requirements:

'This work was supported by a research-promoting grant from Keimyung University Dongsan Medical Center in 2018.'

'The authors received no specific funding for this work.'

Reviewers' comments:

Reviewer's Responses to Questions

**Comments to the Author**

1. Is the manuscript technically sound, and do the data support the conclusions?

Reviewer #1: Yes

Reviewer #2: Partly

2. Has the statistical analysis been performed appropriately and rigorously? 

Reviewer #1: Yes

Reviewer #2: I Don't Know

3. Have the authors made all data underlying the findings in their manuscript fully available?

Reviewer #1: Yes

Reviewer #2: Yes

4. Is the manuscript presented in an intelligible fashion and written in standard English?

Reviewer #1: Yes

Reviewer #2: Yes

5. Review Comments to the Author

Reviewer #1: The authors presented an original article entitled "Analysis of astigmatism outcomes after horizontal rectus muscle surgery in subjects with intermittent exotropia". This study examined the factors affecting corneal curvature change after lateral rectus

recession and medial rectus resection surgery (R&R) in patients with intermittent exotropia (IXT). This was a retrospective cross-sectional study.

My comments to the authors are:

1-In the abstract section line 30 the authors said "The factors found to affect corneal curvature change were sex, age,

extent of surgery, and axial length (AL). I suggest that age is removed as the authors will mention later that age had no effect on the corneal curvature changes, lines 40-41.

2-In the methodology section the authors mentioned that the patients were divided into 2 groups (below 8 years old

and 8 years or older, can the authors explain why in particular, did they choose this age to divide their patients?

3-In the methodology section the authors mentioned that the axial lengths were divided into 2 groups (less than 23.15 mm and 23.15 mm or longer, also no explanation was given for choosing this axial length in particular.

4-In TABLE 1: "Extent of surgery ( [5.0, 4.0], [6.0, 5.0] : above", this sentence is not clear at all, if this is the surgical dose this should be written in a clearer way.

5- In the surgery analytical method: the authors divided the patients into 2 groups (recession : resection ; 5.0 mm : 4.0 mm + 6.0 mm : 5.0 mm versus above, they didn't mention the preoperative angle of deviation and how did they calculate the surgical dose.

6- Authors have to describe in details the difference between the different ways of analyzing K-readings, astigmatism power and axis in the methodology section to be able to understand the difference between ray tracing and simulated mode.

7-In the results section:it is preferable to mention astigmatism in terms of power in diopters and axis in degrees, whenever they are mentioned in the entire manuscript.

Reviewer #2: Good work really but:

1- regarding to amount of surgery we don't know about the higher limit of RR surgery and the average of recession and resection in every subset

2- It was to be very useful if you tried to inspect the relation between the change in astigmatism and the axis of the original astigmatism, this may be more important than the change of the axis (for example; if the positive axis was 75 or 80 will this make the final increase of astigmatism less than that if the axis was 90?)

3- since that maximum age was just 28y and most of patients were around the average age (9.5y), the paper can't conclude that age isn't an important factor, there should be a wide range of ages and a considerable number of adults.

4- the myopic shift can't be determined unless one had did cyclorefraction in every stage of the assessment.

5- the myopic shift can't be physiological (line 252) since the period is just 3 months.

6. PLOS authors have the option to publish the peer review history of their article (what does this mean?). If published, this will include your full peer review and any attached files.

Reviewer #1: No

Reviewer #2: **Yes: **Samer HAJJO

---

## [Author Response · Author response to Decision Letter 0]

15 Sep 2020

Respond to Reviewers has been submitted as a separate file.

---

## [Editor Report · Decision Letter 1]

18 Sep 2020

Analysis of astigmatism outcomes after horizontal rectus muscle surgery in patients with intermittent exotropia

PONE-D-20-20795R1

Dear Dr. Lee,

We’re pleased to inform you that your manuscript has been judged scientifically suitable for publication and will be formally accepted for publication once it meets all outstanding technical requirements.

Kind regards,

Ahmed Awadein, MD, Ph.D, FRCS

Academic Editor

PLOS ONE
---

## [Editor Report · Acceptance letter]

28 Sep 2020

PONE-D-20-20795R1 

Analysis of astigmatism outcomes after horizontal rectus muscle surgery in patients with intermittent exotropia 

Dear Dr. Lee:

I'm pleased to inform you that your manuscript has been deemed suitable for publication in PLOS ONE. Congratulations! Your manuscript is now with our production department. 

Kind regards, 

on behalf of

Dr. Ahmed Awadein 

Academic Editor

PLOS ONE